# Science Across Languages: Assessing LLM Multilingual Translation of Scientific Papers

## Abstract

Scientific research is inherently global. However, the vast majority of academic journals are published exclusively in English, creating barriers for non-native-English-speaking researchers. In this study, we leverage large language models (LLMs) to translate published scientific articles while preserving their native JATS XML formatting, thereby developing a practical, automated approach for implementation by academic journals. Using our approach, we translate articles across multiple scientific disciplines into 28 languages. To evaluate translation accuracy, we introduce a novel question-and-answer (QA) benchmarking method, in which an LLM generates comprehension-based questions from the original text and then answers them based on the translated text. Our benchmark results show an average performance of 95.9%, showing that the key scientific details are accurately conveyed. In a user study, we translate the scientific papers of 15 researchers into their native languages, finding that the authors consistently found the translations to accurately capture the original information in their articles. Interestingly, a third of the authors found many technical terms "overtranslated," expressing a preference to keep terminology more familiar in English untranslated. Finally, we demonstrate how in-context learning techniques can be used to align translations with domain-specific preferences such as mitigating overtranslation, highlighting the adaptability and utility of LLM-driven scientific translation. The code and translated articles are available at this https URL.

## 1 Introduction

Around 98% of all peer-reviewed scientific articles are published in English, but only around 7% of the world's population speaks English as a first language (Liu, 2017). While having a common language among academic journals facilitates international scientific discourse, it also creates a significant barrier to access scientific knowledge for non-native English speakers. For instance, a large-scale survey found that 96% of respondents agree or strongly agree that English as the dominant academic language disproportionately advantages native speakers, among other similar studies (Ferguson et al., 2011) (Tardy, 2004) (Flowerdew, 1999). This linguistic dominance introduces challenges across multiple aspects of science, from biases in peer review against non-native English writers to global implications for science-informed policy (Steigerwald et al., 2022). At the heart of this issue is language accessibility in existing scientific literature. Academic journals, especially widely-read and open-access journals, cater to a global audience (Nature Index, 2024). The availability of scientific literature in a person's native language could play a crucial role in shaping their decision to pursue a career in science.

Many studies have explored potential solutions and paradigm shifts to overcome the language barrier in academic publishing. Given the systemic role that academic journals play in disseminating scientific knowledge, the most direct and impactful solution is for journals themselves to support translations of their articles to other languages. Several challenges hinder the adoption of multilingual translations for journal articles, including cost, logistical complexity, and the question of how best to translate scientific text. The search for a feasible, accurate, and easily adoptable method for translating scientific articles remains elusive.

Machine translation offers a cost-effective and scalable solution for translating text. With the rapid development of neural-based approaches and deep learning, machine translation improved enormously, and neural machine translation (NMT) systems like Google Translate and DeepL have been the gold standard for both general and professional translation tasks (Kalchbrenner & Blunsom, 2013) (Stahlberg, 2019). Recently, with the rise of transformer-based large language models (LLMs), the landscape is shifting. Recent studies show that LLMs match and often surpass NMT systems in performance across a wide variety of translation tasks, including scientific text (Hendy et al., 2023) (Jiao et al., 2023) (Mohsen, 2024). What truly sets LLMs apart, however, is their ability to process complex instructions and generate context-aware, customized outputs. By leveraging simple in-context learning techniques alone, LLMs can be trained to produce translations that are specifically tailored to the requirements of the scientific community, accommodating factors like formatting preferences and domain-specific vocabulary. This flexibility opens the door to incorporating a wide range of potential feedback from non-native English-speaking researchers, enabling more specialized and effective translations.

In this article, we develop LLM-backed automated translation solutions to support lowering the language barrier in the scientific community. We introduce a method for generating publisher-ready full-length article translations, propose a novel QA benchmarking strategy to evaluate translation quality, and demonstrate how LLM few-shot prompting can be used to integrate feedback from actual authors of research papers into the translation process. We assess the strengths and weaknesses of LLM translation through both automated evaluations and user studies.

## 1.1 Related works

Several studies have evaluated the performance of LLMs (e.g. GPT models) on various translation tasks, showing that many are competitive with previous state of the art NMT systems, especially more recent models such as GPT-4 (Hendy et al., 2023) (Jiao et al., 2023). Further developments in LLM-based translation include prompting techniques (Vilar et al., 2022) (Zhang et al., 2023), context aware and document-level translation (Wang et al., 2023), translations that adapt to user feedback in real time (Moslem et al., 2023a), non-English monolingual corpora fine-tuning (Xu et al., 2023), and fine-tuning to emulate professional human translation strategies such as analyzing specific parts of a sentence before translating (He et al., 2024).

When it comes to assessing machine translation of scientific journal articles, the literature is more sparse. Zulfiqar et al. (2018) applied a variety of NMT systems, including Google Translate and DeepL, to translate excerpts of German scientific articles from the last century. Other studies focused on scientific abstracts (Tongpoon-Patanasorn & Griffith, 2020) (Wei, 2017). To the best of our knowledge, all other studies on scientific translation were specialized to the medical field (Soto et al., 2019) (Daniele, 2019) (Sebo & de Lucia, 2024). Although the topic of translating full-length academic journal articles has yet to be thoroughly investigated, many studies have introduced general LLM-backed translation strategies for technical and terminology-heavy text. Some of those strategies include term extraction and glossary creation (Kim et al., 2024), RAG-based dictionary retrieval (Zheng et al., 2024), and using LLM-generated synthetic data to train proper usage of domain terminology (Moslem et al., 2023b).

The most widely used and convenient methods for benchmarking machine translation are automated metrics such as BLEU (Papineni et al., 2002), BLEURT (Sellam et al., 2020), ChrF (Popović, 2015), TER (Snover et al., 2006), and COMET (Rei et al., 2020). These metrics are typically applied to source-target translation pairs from established datasets from the Workshop on Machine Translation (WMT) or FLoRes (Goyal et al., 2022) (for low resource languages). The vast majority of these metrics require established reference translations, with the notable exception of COMET-Kiwi (Rei et al., 2022). Similarly, most metrics are trained on (if model-based) and/or evaluated at the sentence level only, with the exception of the document-level (but not reference-free) $d$-COMET (Vernikos et al., 2022). In addition to these automatic metrics, human evaluation is often employed to provide a more nuanced and reliable assessment of translation quality. Parallel datasets have a few drawbacks, primarily that they contain a limited number of language pairs and are restricted to specific topics. When it comes to scientific text, WMT offers parallel biomedical datasets (Neves et al., 2022), but none for scientific/academic text at large scale.

Pengpun et al. (2024) implemented a No Language Left Behind (NLLB) (Costa-Jussà et al., 2022) model that supports code-switching (keeping some terminology in English) in Thai-English medical translation, constituting the only study to our knowledge that fine-tunes the translation to an established preference of end-users (in their case, medical physicians). Another study analyzed research abstracts from English and Chinese articles and found substantial differences in rhetorical conventions (Li, 2020). We were not able to find systematic studies on the preference of researchers on academic translation.

### 1.2 Our contributions

**Journal-compatible translation.** To the best of our knowledge, we develop the first pipeline using LLMs to translate scientific articles while preserving standard publishing formats (JATS XML). We produce a total of 903 full translations throughout the study:

- 28 languages × 6 articles = 168 translations for the QA benchmark in Section 3.

- 15 translations for the user study in Section 4.

- 10 languages × 24 articles × 3 models = 720 translations for the multi-model QA benchmark in Section A.1.1.

**Automated QA benchmarking.** We have developed an automated benchmarking method specifically tailored for scientific documents (Section 3). Whereas Krubiński et al. (2021) propose a QA technique for evaluating sentence-level machine translation against a curated set of (mostly to-English) reference translations, here we show that QA methods are also useful to evaluate document-level translations without references. Our approach requires only the original and translated documents to assess translation quality, making it language agnostic, article specific, and independent of parallel translation datasets.

**Translation preferences for scientific text.** We have gathered feedback on machine translations in a variety of languages directly from authors of research papers across multiple scientific disciplines. Subsequently, similar to Pengpun et al. (2024), we have also created code-switched translations; instead of masking, we implemented few-shot prompting using a scientist-curated example translation.

## 2 Journal-compatible translation

Journals have the power to change language barrier norms, as they serve as the primary forum for scientific knowledge. However, for multilingual translation to be widely adopted in scientific publishing, the process must be practical for journals to implement. In this section, we demonstrate how LLMs can preserve the formatting of journal articles during translation, offering an approach that is adaptable and easy to integrate.

In 2002, the NIH introduced the Journal Article Tag Suite (JATS), an XML protocol for structuring scientific journal articles. Since then, JATS has become part of the National Information Standards Organization (NISO) and is the global standard for academic publishing. Despite the distinct "look and feel" of articles across different publishers, they all share the same underlying JATS XML structure. For instance, academic journal articles universally include <front>, <body>, and <back> sections that contain the main text; <article-title> and <abstract> sections; a <contrib-group> section that stores author information, and many more (Needleman, 2012).

We employ an LLM (GPT-4o) to translate journal articles in their native JATS form, ensuring that the XML structure remains intact while translating the content. Figure 1 illustrates the core principal of the approach. When tasked with translating this section ("Sec5"), GPT-4o successfully translates the text while preserving the surrounding <sec>, <title>, <p>, and <xref> tags. A full article is much more complex, however, consisting of multiple, heavily nested elements that include figures, tables, equations, and more. We translate each full article in a series of API calls to GPT-4o; we find that processing more than roughly 5 paragraphs at a time occasionally results in truncated translations. To increase context awareness, we prepend the prompt with the contents of the full original document.

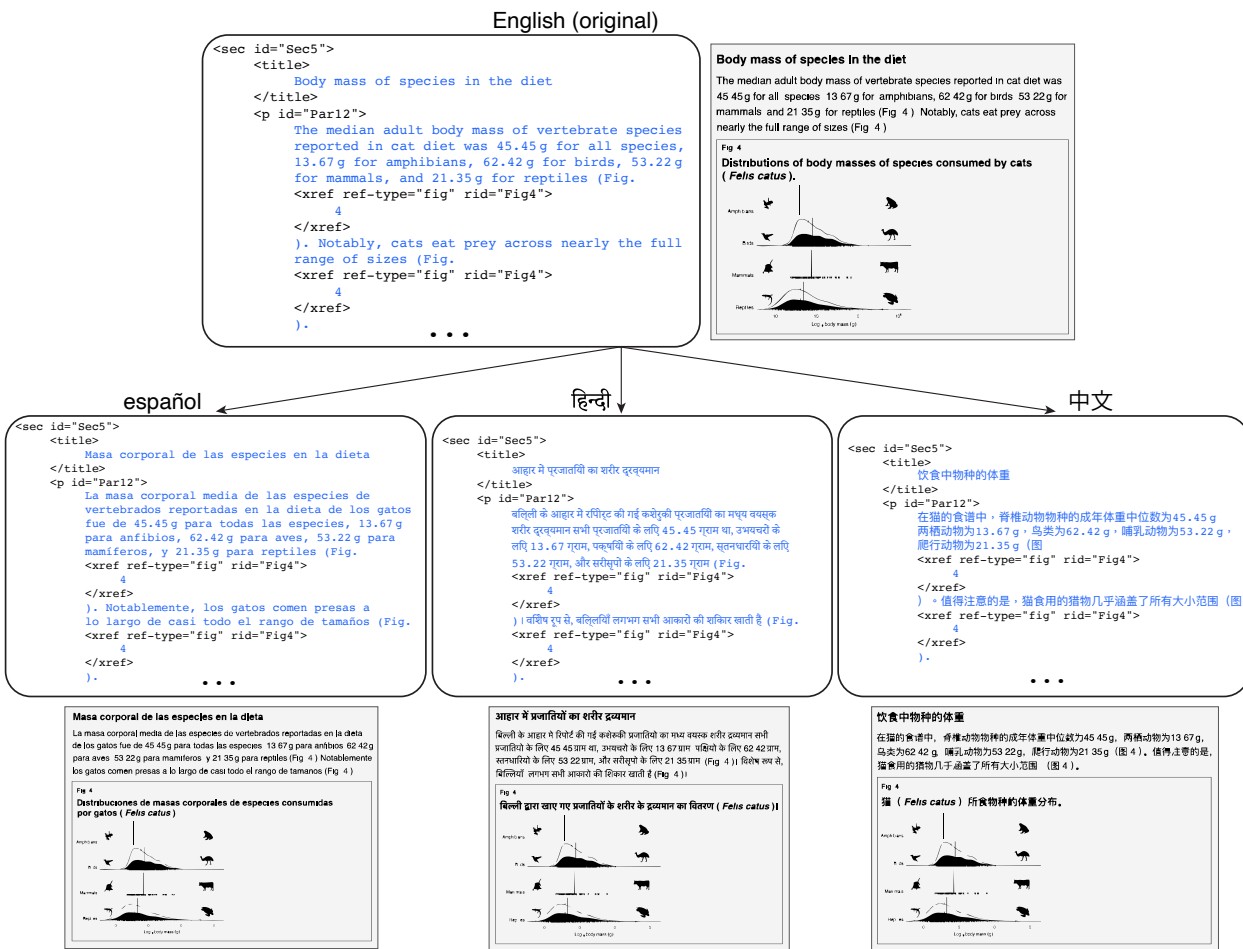

Figure 1: Example of a JATS-formatted article snippet (Lepczyk et al., 2023), translated with our method into three of the 28 languages included in our study (Spanish, Hindi, and Chinese). The XML tags (black) are preserved while the article text (blue) is translated. Below each translated JATS XML snippet is the resulting section of the HTML-displayed article.

Even for complex elements, we find that GPT-4o reliably maintains XML formatting without introducing errors. However, occasional issues arise with nesting, such as paragraph text incorrectly appearing inside a figure caption. To ensure structural accuracy, we translate tables and figures—typically the most complex elements in a JATS article—independently before appending them to their respective sections. Additionally, we observe that special characters like '<' and '>' can sometimes cause truncation of the article text, even though the XML structure remains intact. To address these issues, we modify the prompt to include *"Do not cut sentences short and include all symbols,"* after which the model produces fully translated articles with very few structural issues. In the 408 translations we generated with GPT-4o and QA-evaluated over the course of this study, we identified only two with truncation errors and six with nesting errors, resulting in a 98% accuracy in preserving the original JATS structures

Using our method, we successfully translate full articles into 28 different languages while fully preserving the JATS formatting. Because of its compatibility with native article formatting, this translation step can be applied at the final stage of publication or to articles that have already been published. This ensures maximum compatibility with the publication framework and paves the way for widespread translation of articles across different journals. Moreover, the translation is applicable to an arbitrary number of languages. While JATS is the ubiquitous standard, this approach is adaptable to other XML protocols as long as the tag suite is properly documented, ensuring broad compatibility across scientific publishing. A database of

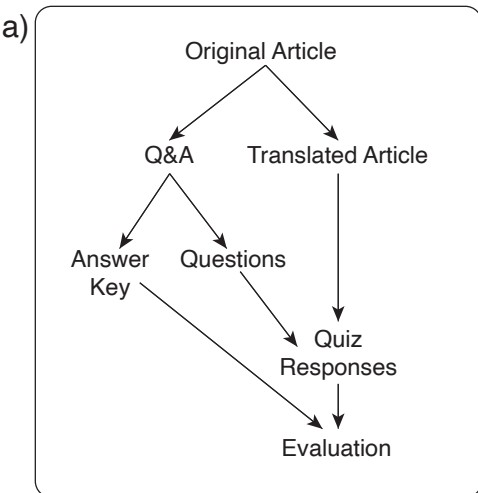
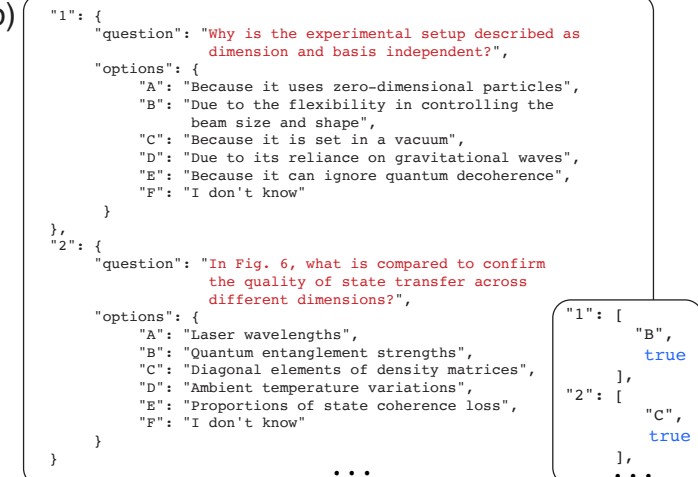

Figure 2: QA benchmark process. **a.** Flowchart schematic of the benchmark process. **b.** An example of two quiz questions generated from a scientific article. In this case, the model reading the translated article answered both questions correctly (inset), indicating that the scientific details covered by those questions were accurately translated.

all GPT-4o translated articles in this study, totaling to 423 translations, is available on our webpage: https URL.

## 3 QA-style automated benchmarking

In this section, we evaluate the translation quality of our approach. Traditional machine translation evaluation relies on automated benchmarking metrics such as BLEU, which compare translations against parallel reference data. However, to our knowledge, no dataset exists that provides parallel, document-level scientific translations across the diverse range of languages and disciplines we have included here. Instead, we introduce a novel question-and-answer (QA) style benchmarking method. In this approach, an LLM generates a "quiz" with multiple-choice questions designed to capture key details from the original scientific article. The LLM then "reads" the translated article and attempts to answer these questions based solely on the translated content. The higher the accuracy, the better the translation conveys the scientific details of the original text.

A key advantage of this benchmarking method is its automation and adaptability. Unlike traditional evaluation techniques, it does not require parallel translation data and is therefore applicable to any article, in any format, and in any language. This flexibility is particularly valuable for evaluating translations in "low-resource" languages, where high-quality parallel datasets are scarce.

### 3.1 Benchmarking procedure

The benchmarking procedure is illustrated in Figure 2. First, we prepare the quiz by providing GPT-4o with the original English text and prompting it to create 50 multiple-choice questions that encapsulate key details of the paper, along with a corresponding answer key. The exact prompt is as follows:

*"Please read the following scientific journal article. Generate 50 detailed and specific questions to test a reader's understanding of the findings of the article. Each question should be unique. The questions should labeled 1-50. The questions should be multiple choice with 6 possible answers: 5 are labeled A-E, and the 6th option should say 'I don't know'. There should only be one correct answer from the options. The questions should cover the unique results, figures, and tables of the article as much as possible."*

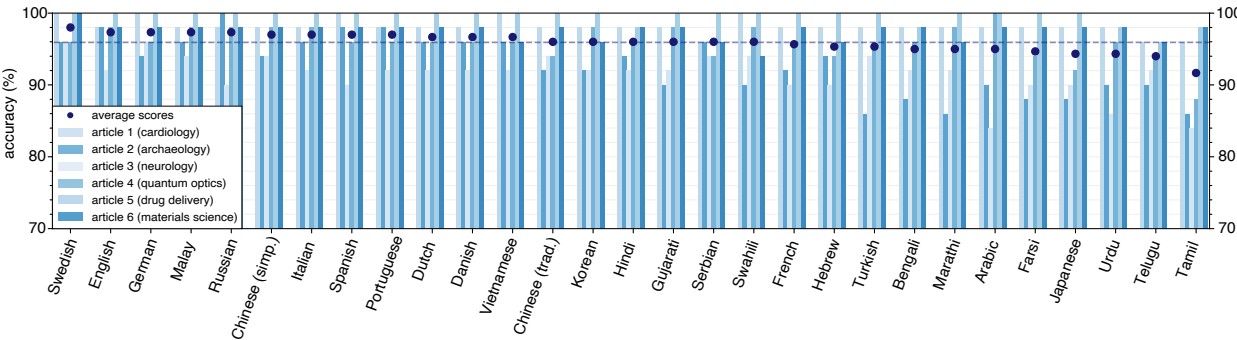

Figure 3: QA benchmarking results for six articles translated into 28 different languages, plotted by highest average score. The dashed line indicates the overall average performance (95.9%) across all languages and articles.

To execute the benchmark, we then prompt the model to read the translated article and answer the quiz questions. In this scenario, the model simulates a real person reading the translated text; if the translated article effectively conveys the core details and central findings, the model should perform well on the evaluation. To better reflect this scenario, we also translate the quiz questions into the target language. This ensures the model reads both the article and the questions in the same language, mirroring how a native speaker would engage with the material. Benchmarking results using untranslated (English) quiz questions are reported in Section A.1.1.

The model's quiz accuracy, graded against the answer key, constitutes the benchmark result. For instance, if the LLM correctly answers 48 out of 50 questions for a given article in a particular language, the benchmark score for that translation would be 96%. To ensure that the quiz-taking LLM relies only on the translated article rather than its prior knowledge, we implement two safeguards. First, we do not include the quiz-generation exchange when prompting the model during the evaluation; the model receives only the translated article and the quiz questions. Second, we prevent pre-training contamination by only including articles that score 0% on the benchmark when the article is not provided, i.e. the model selects "I don't know" for all questions. Full details on prompts and model parameters are provided in the Appendix.

### 3.2  Results

For this study, we apply the QA benchmark to six articles spanning a wide range of disciplines, from medicine to archaeology to quantum optics. Each article is translated into 28 different languages, which were selected based on countries in Nature Index's Research Leaders list and further supplemented to include languages from more regions of the world. Then all 28 languages × 6 articles = 168 translations are evaluated with the QA benchmark. We also include an English baseline, which we perform by conducting the quiz on the original article. Figure 3 presents the results. The overall average performance across all 29 languages and all six articles is 95.9%, with the lowest average score at 91.7% (Tamil) and the highest average score at 98.0% (Swedish). Notably, no individual quiz score falls below 84%, and translations in 23 languages score 100% on at least one article. These high QA results indicate that our translation approach effectively conveys the key findings and essential details of scientific articles across diverse disciplines.

The English baseline score (97.3%) is higher than the overall average, but not a perfect 100%. We attribute this to two potential factors: (1) the quiz-taking LLM, like a human reader, may exhibit minor imperfections in reading comprehension, leading to occasional errors even when working with English text or high-quality translations; and (2) quiz questions or answer choices may be occasionally ambiguous (see Section A.1.4 for further analysis). In a broader evaluation across 24 papers (Section A.1.1), the English baseline (96.8%) is much closer to the top score (Spanish, 97.0%), suggesting that as the benchmark scales, English is likely to yield the more intuitive outcome of highest average performance overall. While refining the quiz questions could further improve the benchmark, we believe the current methodology already provides a reliable evaluation framework.

Additionally, our results reveal that "low-resource" languages such as Urdu, Telugu, and Tamil perform slightly below high-resource languages, aligning with prior findings in both machine translation and multilingual LLM research (Nicholas & Bhatia, 2023) (Jiao et al., 2023). However, since even the lowest-performing languages achieve an average accuracy above 91%, this effect is minor, demonstrating that our benchmarking technique is applicable across a wide range of languages.

Comparisons with the same articles translated as plain text (without JATS formatting) by GPT-4o and Google Translate further support our finding that highly structured text is translated just as effectively as unstructured text. We observe no degradation in translation quality due to the additional task of preserving structured content. Specifically, the average benchmark score for GPT-4o's XML-based translations (95.9%) closely matches that of GPT-4o's plain text translations (96.0%) and Google Translate (95.8%) (Section A.1). While this study focuses on benchmarking JATS-formatted translations as a form of customization, our QA-based evaluation method is broadly applicable for evaluating translations across various formats, even when other types of translation customizations are applied.

## 4   Feedback from authors

In this section we complement the QA benchmarking results with evaluations from 15 human scientists across various languages and disciplines, including theoretical and experimental quantum optics, nanophotonics, biostatistics, materials science, magneto-electronics, machine learning, and more. In this study, each participant is provided with a translation of their own scientific paper in their native language, generated using our method. We then gather feedback on translation quality using the following questions:

1. How effectively does the translation **convey the original information** of the article?

2. How well do you think another speaker of this language would be able to **understand the key ideas** of this paper just from this translation?

3. How satisfied are you with the translation of **technical terms** in the article?

4. How well does the translation **flow and maintain cohesion** throughout the text?

5. How well does the translation maintain the original **tone and style** of the article?

For each question, the three possible options are *few or no issues*, *some issues*, *many issues*, and *other*. Participants also have the opportunity to provide free-form comments with their observations and opinions. Questions 1 and 2 target the accuracy and main details, similar to the QA benchmark, while questions 3, 4, and 5 probe stylistic and subjective aspects of translation quality. Through this questionnaire we aim to gain deeper insights into the academic community's perspective on what defines effective scientific translation.

As expected, nearly all researchers in our study (93.3%) report that the translation of their paper contains *few or no issues* in conveying the original information, reinforcing the findings from our QA benchmarking. Participants also generally agree (86.7%) that other scientists reading their translated paper would understand the key ideas with *few or no issues* (Fig. 4a). Based on some participant comments, the most commonly cited issues in this area include minor misinterpretations and inconsistencies in vocabulary (e.g., a specific word being translated differently throughout the text).

Key insights arise from the more subjective questions. As one might anticipate from machine translation, authors rate lower scores in the categories of tone and style, flow and cohesion, and technical terms. In particular, many participants (86.7%) describe an unnatural quality to the translation or dissatisfaction attributed to the handling of technical and domain-specific vocabulary. With regard to technical vocabulary, participants reported two kinds of issues:

(i) **Mistranslation**: This technical term exists in their native language, but the model translated it awkwardly or incorrectly.

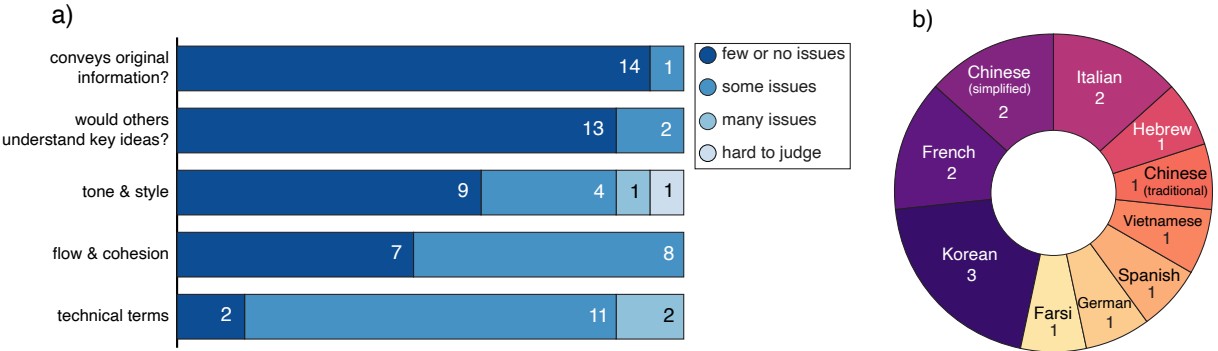

Figure 4: Feedback from scientists. **a.** Survey responses from 15 participants after reading their translated paper, with questions sorted by average score. **b.** Languages represented by the participants.

    (a) **Example 1**: The model translated *edge coupling* into French as *couplage par bord*, but the more commonly-used phrase is *couplage par la tranche*.

    (b) **Example 2**: The model translated *switching* (e.g. magnetic switching) into Chinese as 切换, but 轉換 is a better fit.

  (ii) **Overtranslation**: This technical term does not exist in their native language, or is rarely used in practice, and the original English word is preferred.

    (a) **Example 1**: The model provided a literal translation of *rigorous coupled-wave analysis* into Korean (엄밀 결합 파동 해석), but using the English term is preferred.

    (b) **Example 2**: The model translated *gap* (e.g. Hamiltonian/energetic gap) into Spanish as *brecha* (breach). A better translation might be *salto*, as in *salto de energía* (energy jump), but many scientists would simply use the English *gap*.

Whether certain terms might be more appropriately left untranslated is not a typical factor in traditional machine translation. However, the feedback from scientists highlights the importance of this consideration in scientific translation. The frequency of overtranslation comments in our survey responses (33.3%) suggests the need for nuanced translation approaches that align with how technical terms are used in practice.

## 5 Feedback-adaptive translation

In this section, we leverage LLM output customization to incorporate the feedback from scientists. In particular, since so many scientists expressed a preference for retaining some technical terms in English, we apply a targeted prompting technique to preserve some English vocabulary during translation.

To translate text while maintaining the appropriate English terms, a key challenge is the inherently subjective nature of deciding which terms to keep and which to translate. To navigate this, we employ few-shot prompting, an in-context learning technique where GPT-4o is provided with verified examples to improve responses in scenarios where data is scarce (Brown et al., 2020). Specifically, we construct a one-shot prompt using a translated paragraph from a scientific article in which the author of the article has reviewed the translation and identified terms that should remain in English. This curated example then serves as a guide for translating other texts (Fig. 5a).

Using this prompting method, we generate new translations and seek feedback from five authors who previously expressed concerns about technical term translations. Each participant reviews two versions of an excerpt from their paper: one direct translation and one generated with the one-shot prompt that retains some English terms. They are then asked to indicate their preference between the two versions.

The original excerpts for the five participants in this follow-up study range from approximately 153 to 308 words in length, with an average of 222 words (roughly 1–2 paragraphs). Following translation using the

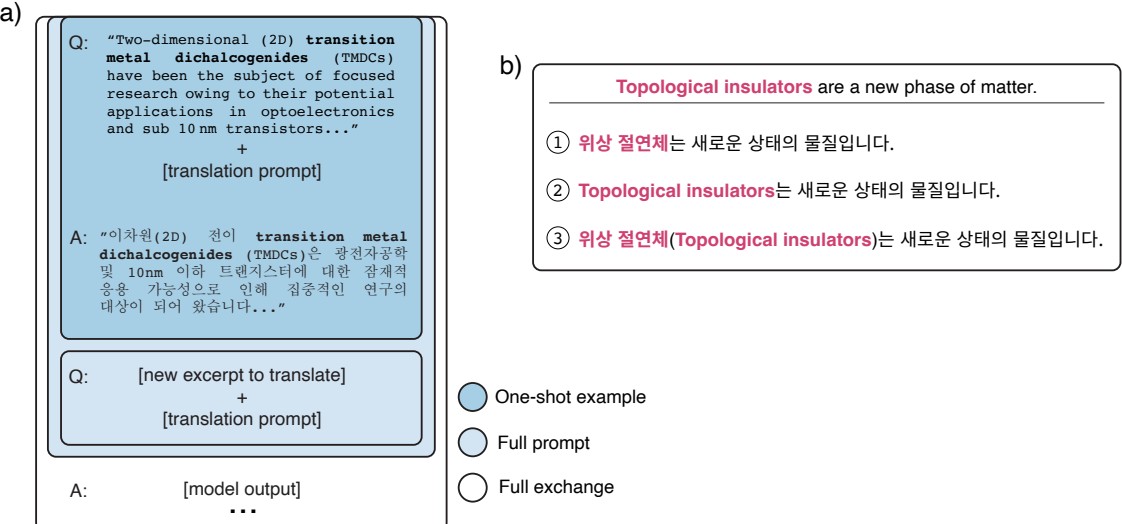

Figure 5: In-context learning to customize scientific translation. **a.** One-shot prompt construction. In the one-shot example, the word "transition metal dichalgocenides" (in bold) is kept in English, while the rest of the excerpt is translated. Full prompts are available in the Appendix. **b.** An example sentence translated three different ways, per feedback from scientists: 1) direct translation, 2) preserving the technical term in the original English, 3) direct translation plus original English word in brackets.

curated one-shot prompt, a manual count reveals that the number of untranslated English terms in the translated texts ranges from 23 to 65 words, averaging 34 words per excerpt. This corresponds to an average English word retention rate of 15.6%, which is comparable to the proportion of English words present in the curated one-shot example (11.4%). In prior translations without the curated prompt, all English terms were consistently translated into the target language, indicating that the preserved English words in these translated excerpts are a direct result of the one-shot prompt's influence. Notably, the translated excerpt with the highest rate of English word preservation (34.2%) suggests that the model is capable of adapting to scientific texts containing a higher density of technical terminology.

The results of the follow-up survey reveal a diverse range of preferences. As anticipated from the initial survey, three of five participants find that retaining some English terms produces a more natural and readable scientific text. Conversely, the other two participants are more inclined toward the complete translation, citing a preference for better-translated terms over English terms. From the responses, one interesting observation is that speakers of languages with a higher prevalence of English loanwords, such as Korean, tend to favor English technical terms compared to those from languages with fewer English loanwords, such as French, a phenomenon which might be influenced by historical linguistic reasons[1] (Blackwood, 2013) (Tyson, 1993). Additionally, one participant proposes a balanced approach: to present the original English term in brackets alongside the translated word, rather than strictly choosing one over the other (Fig. 5b). The strength of LLM-based translation lies in its ability to integrate diverse customization and feedback, enabling tailored and therefore more effective translations. While this study focuses on the overtranslation phenomenon, the prompting technique we utilize in this section can be applied further to other vocabulary or stylistic preferences by incorporating additional examples.

## 6 Discussion

In this study, we utilized LLM-powered translation to go beyond traditional plain-text translation, resulting in scientific translations that are tailored with both publishers and authors in mind. Our method successfully translates scientific articles in JATS XML while maintaining the complex structure, opening a new avenue

---

[1]For instance, many scientific terms in English originally derive from French (Faure, 2018).

for academic journals to include translations for their articles. Through a novel automated QA benchmarking approach, we quantitatively evaluated full article translations across 28 languages and many scientific disciplines, finding that key scientific details are reliably conveyed even in low-resource languages. Further, our human evaluation study revealed valuable insights into the qualitative aspects of scientific translation. While the survey results confirmed the high overall translation accuracy, they also highlighted areas for improvement, such as the handling of technical vocabulary. By leveraging the few-shot prompting technique, we incorporated feedback and generated customized translations that align with the linguistic preferences of researchers across different fields and languages.

Ultimately, our findings emphasize that the flexibility of LLMs allows for nearly limitless degrees of customizability, making it possible to improve translations based on domain-specific requirements and preferences. This adaptability presents a significant step toward breaking language barriers in academic publishing, fostering broader accessibility and collaboration in global research.

## 6.1 Comparison of models

All of the translations and QA evaluations detailed in this main text were performed with GPT-4o-2024-08-06. We also evaluated Llama 3.3 Turbo Instruct and Qwen 2.5 Turbo Instruct for the translation methods introduced in this study, but found both models less suitable in comparison to GPT-4o. Specifically, both models were more prone to truncation errors, frequently omitting sections of the article (e.g. tables, figures, and equations) and resulting in less comprehensible translations. For a full analysis, see Section A.1.1.

## 6.2 Limitations and future directions

Two participants in our user study reported inconsistencies in the translation of certain terms throughout the article. This likely stems from our approach of translating articles in separate sections to mitigate truncation and XML nesting issues (Section 2), leading to potential variations in the model's vocabulary choices between different API calls. One possible solution is to track all translations of the same term and standardize them at the end by replacing inconsistent terms with the most common translation. Furthermore, while our method includes the full original article in the prompt to provide context, further research could explore ways to enhance context-awareness in scientific translation, which may also help reduce vocabulary inconsistencies. Additionally, it is possible that similar translation quality could be achieved using only the local context surrounding the section being translated, instead of the entire article, which would substantially reduce the number of tokens required per translation. Future research could explore how much context is necessary for accurately translating subsections of scientific text, potentially enabling more efficient translation.

In addition to the QA-based evaluation proposed in this study, it would be valuable to incorporate more traditional translation metrics such as BLEU or COMET. While most of these automated metrics rely on reference translations and are therefore not directly applicable to our setting, devising a reference-free metric (like combining COMET-Kiwi with $d$-COMET) could serve as a useful complement to our QA benchmark.

Future work could also expand on understanding researchers' preferences in scientific translation. A larger-scale survey may reveal additional patterns in translation preferences, particularly across different languages (as discussed in Section 5). Moreover, if our method or a similar approach is adopted for large-scale scientific translation, the resulting corpora could be used to fine-tune models, further improving consistency and overall translation quality.

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

# A   Appendix

## A.1   QA benchmark

The journal articles translated for the study in Section 3:

- Article snippet in Figure 1: Lepczyk et al. (2023)
- Article 1: Heo et al. (2024) (13,413 tokens)
- Article 2: Fernández-Crespo et al. (2023) (25,057 tokens)
- Article 3: Rus et al. (2023) (26,973 tokens)
- Article 4: Sephton et al. (2023) (56,845 tokens)
- Article 5: Yang et al. (2023) (24,317 tokens)
- Article 6: Peng et al. (2023) (14,881 tokens)

Model information for generating and executing the QA benchmark:

- **Model**: gpt-4o-2024-08-06
- **Temperature**: 1
- **Quiz generation prompt:** "Please read the following scientific journal article. Generate 50 detailed and specific questions to test a reader's understanding of the findings of the article. Each question should be unique. The questions should labeled 1-50. The questions should be multiple choice with 6 possible answers: 5 are labeled A-E, and the 6th option should say 'I don't know'. There should only be one correct answer from the options. The questions should cover the unique results, figures, and tables of the article as much as possible. If you are able to answer any of the questions without having read the article, please generate a better question. Please format your response as a JSON object with the question, possible answers, and correct answers. The JSON key to each question should be its number. Here is the article: [*original article*]"
- **Quiz execution prompt**: "Please read the following scientific journal article, which has been translated into [*lang*]. Then answer the questions based on your understanding. Report your answers as a JSON where the keys are the question numbers and the values are your letter answers. Here is the article to read: [*translated article*] and here are the questions: [*questions*]. If you do not know the answer, select 'I don't know' as your answer. Do not make guesses."

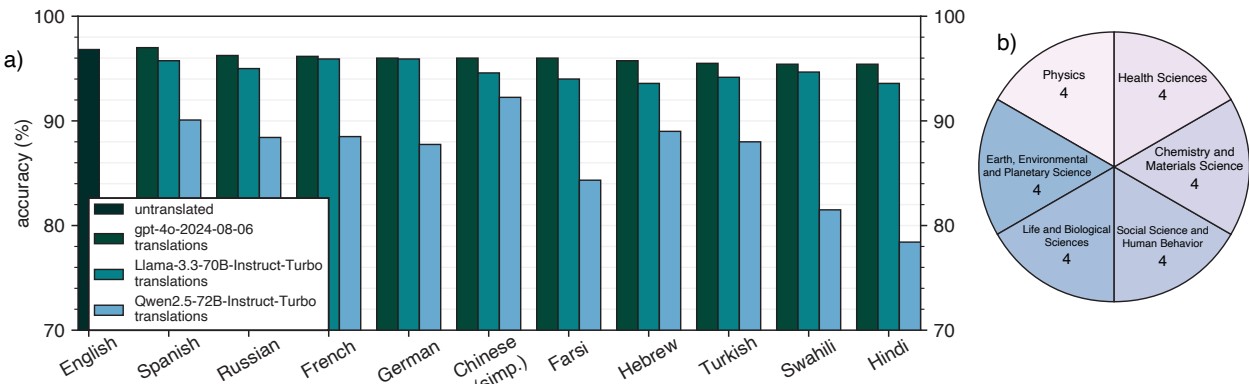

Figure 6: Extended QA benchmarking results, using 24 articles, 10 languages of translation, and three different models for translating – totaling to 24×10×3=720 translations evaluated (excluding the English baseline). **a.** Excluding English, the average performance for GPT-4o, Llama-3.3, and Qwen-2.5 translated articles is 95.5%, 94.7%, and 86.6%, respectively. **b.** The 24 articles translated are sourced evenly from the six categories of *Nature Communcations* Top Articles of 2024.

We select temperature 1 for the question generation because we want the QA to include as many questions as possible. For temperatures below 1, we notice repeated questions when generating 50-question sets, whereas the 50 questions generated with temperature 1 are always distinct. For temperatures above 1, we observe hallucinations, i.e. nonsensical outputs.

### A.1.1 Comparison of models for translation

In this section we investigate an additional two large language models, Llama-3.3-70B-Instruct-Turbo (Grattafiori et al., 2024) and Qwen2.5-72B-Instruct-Turbo (Yang et al., 2025). For this study, we select 24 articles, with four articles from each of the six categories of the *Nature Communications* Top Articles of 2024 (Health Sciences; Life and Biological Sciences; Social Science and Human Behavior; Chemistry and Materials Science; Earth, Environmental and Planetary Sciences; Physics). These articles are distinct from the six used in the single-model study detailed in Section 3. The 24 articles are each translated into 10 languages, selected as a representative subset of the original 28 to reduce computational costs while preserving cultural diversity and broad global coverage. After confirming the absence of pretraining contamination, we execute the benchmark (with translated quiz questions) on all 24 articles × 10 languages × 3 models = 720 translations, as well as the English baseline. The untranslated articles are 26,507 tokens on average.

The benchmark results are presented in Figure 6. While the GPT-4o results are comparable to the six-article study in Section 3, the other models consistently perform worse, especially the Qwen-2.5 model. Upon analysis, it appears that the sizable difference in benchmark performance is predominantly due to truncation errors, i.e. the model omitting parts of the XML chunks. Across the 240 translations generated by each model, we evaluate whether any equations, tables, or formulas are missing. Only 2 translations from GPT-4o omit such elements (specifically, both are missing an equation), compared to 53 translations from Llama-3.3 (22%) and 168 from Qwen-2.5 (70%). The subsequent lower performance of Llama-3.3 and Qwen-2.5 translations on the benchmark is consistent with such errors, since some benchmark questions specifically reference information from tables and figure captions. Since equations and tables are often among the most deeply nested elements in the XML structure, it is possible that Llama-3.3 and Qwen-2.5 may also be omitting other content (e.g. paragraphs containing 'lists', 'blocks' or other unaccounted-for elements). This issue could potentially affect GPT-4o as well; however, its high benchmark performance suggests that any such omissions, if present, do not result in a meaningful loss of information—unlike the gaps observed in the Llama-3.3 and Qwen-2.5 translations.

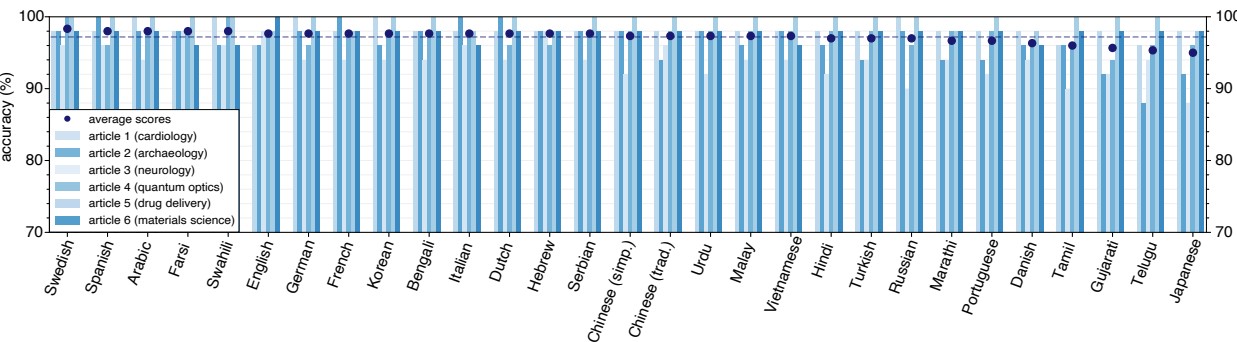

Figure 7: QA benchmarking results, where quiz questions are kept in English, plotted by highest average score. The dashed line indicates the overall average performance (97.2%) across all languages and articles.

The Qwen-2.5 translations also exhibit greater variability in benchmark scores across languages, potentially suggesting increased sensitivity to lower-resource languages. Hindi and Swahili are among the lowest-performing, while the Chinese score is almost on par with Llama-3.3, possibly reflecting stronger capabilities in higher-resource languages. However, since this variability appears to be at least partially due to missing elements in the translations, it remains unclear to what extent language resource levels are actually driving these performance differences.

Overall, we find that GPT-4o is the most suitable model for translating scientific articles in JATS XML format. Future work could extend this benchmark to include a broader range of models, e.g. translation-specialized models, for an even more comprehensive comparison.

### A.1.2  Translating quiz questions

In generating the data in Figure 3 of the main text, we translate the quiz questions into the target language before executing the quiz benchmark. We translate the questions with the following prompt (temperature = 0):

- "The following JSON comprises a list of questions about an academic journal article. Please translate the questions and options into [*lang*]. Do not translate the keys of the JSON. Please return the translated JSON. Here is the JSON to translate: [*questions*]"

Here, we perform the quiz benchmark without translating the quiz questions (keeping them in English) and find an increase in overall accuracy from 95.9% to 97.2% (Fig. 7), suggesting that the language of the quiz questions may play a role in the model's general performance. Whereas in Figure 3 the model emulates a human reader engaging with both the article and QA questions in their native language, here the model is removed from that scenario and simply operating in its strongest and most-aligned language, hence the increase in performance.

### A.1.3  Comparisons with plain text translations

We translate the six articles using the same model (GPT-4o-2024-08-06) but processed the article as plain text, not JATS-formatted, as well as with Google Translate (GNMT) and compare the benchmarking results. The performance among all three translation methods were very similar (Fig. 8), indicating that our JATS-formatted translation method sees no degradation as a result of the LLM parsing XML at the same time as translating.

### A.1.4  Quiz questions with high error rates

In Figure 9 we analyze the number of incorrect responses for specific quiz questions across all six articles and 29 languages. In particular, we note that question # 8 on Article 3 was incorrect for all languages, including

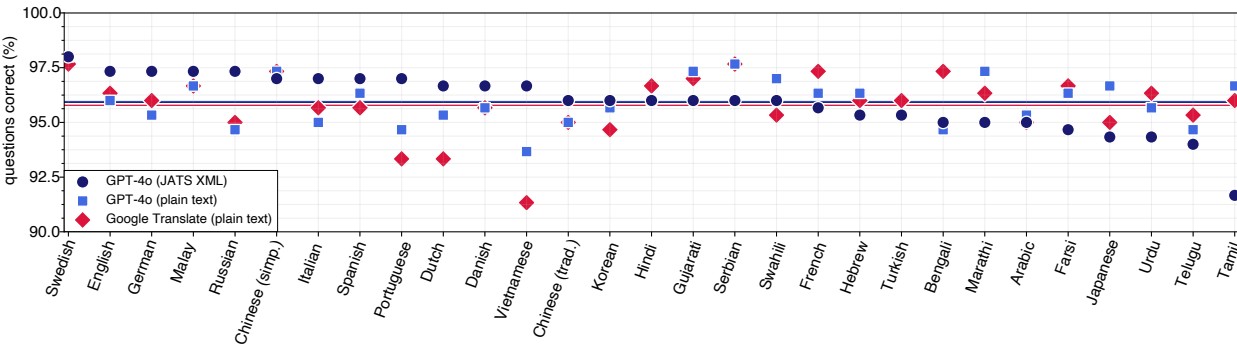

Figure 8: Our XML-based translation approach benchmarked against plain text translation by GPT-4o and Google Translate (GNMT). Solid lines represent the average benchmark scores: 95.9% for our XML-based approach, 96.0% for GPT-4o plain text translations, and 95.8% for Google Translate.

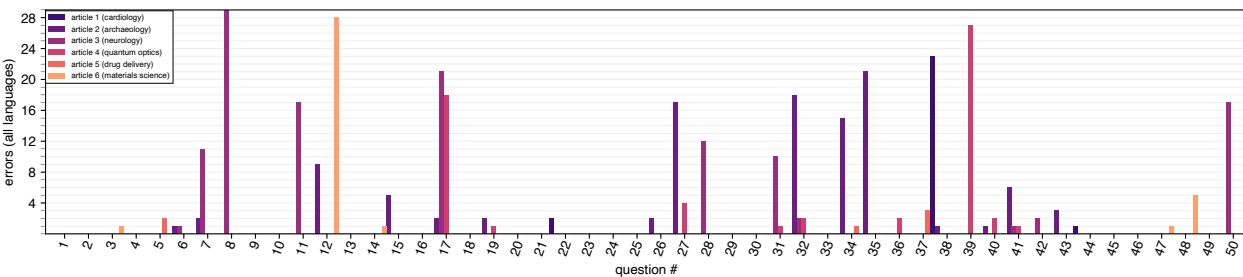

Figure 9: Number of incorrect responses on the QA benchmark for each quiz question (1-50).

English. The quiz question is as follows: "What percentage of patients reported dissociative symptoms that disappeared after SSRI treatment?" with possible answers "15%," "25%," "35%," "45%," "55%," and "I don't know." Upon investigation of Article 3, the study reports that all patients who experienced dissociative symptoms before SSRI treatment had those symptoms alleviated with SSRIs. The quiz question is therefore malformed, and further research could be useful to determine methods for generating more robust quiz questions.

## A.2   One-shot prompt for technical terms

Here we provide the full prompt used for translating excerpts while preserving some English terminology. First, we describe the one-shot example prompt:

**Q:**   "Here is an excerpt of a scientific article: [*original text*]. Please take note of any highly domain specific words in this excerpt. Then, please translate the excerpt into Korean. But do not translate those highly domain specific words that you identified. For those words, keep the original English words in your translation instead. Everything else in the excerpt should be translated into Korean."

**A:**   [*translated text with some technical terms preserved*]

We use the following excerpt from Ahn et al. (2017) as our author-curated one-shot example:

**Original excerpt:**   *Two-dimensional (2D) transition metal dichalcogenides (TMDCs) have been the subject of focused research owing to their potential applications in optoelectronics and sub 10nm transistors. The primary attraction of TMDCs such as MoS2 and WSe2 for both applications is their naturally terminated surface, which allows them to be scaled down to the atomic limit without the concern of surface dangling bonds. Furthermore, in many 2D materials, a number of desirable properties emerge at the monolayer limit,*

*the most notable of which being the presence of a direct bandgap. Many studies based on mechanical bending of exfoliated 2D TMDCs have been conducted on flexible substrates, and they have shown that the application of strain can tune the properties of this new class of materials. For example, it has been demonstrated that in multilayer WSe2, particularly in nominally indirect-gap bilayer WSe2, application of tensile strain can result in a transition from an indirect-to-direct bandgap. Growth on epitaxial substrates with a controlled lattice constant mismatch has typically been utilized to establish built-in strain in three-dimensional semiconductors. However, due to the relatively weak interaction between 2D materials and substrates, this established method of strain engineering is likely not applicable for the strain-engineered growth of TMDCs. In this work, we demonstrate strain engineering of 2D materials directly via chemical vapor deposition (CVD) growth while simultaneously maintaining high material quality, by utilizing the thermal coefficient of expansion (TCE) mismatch between the TMDC and the growth substrate.*

**Translation:** *이차원(2D) 전이 **transition metal dichalcogenides (TMDCs)**은 광전자공학 및 10nm 이하 트랜지스터에 대한 잠재적 응용 가능성으로 인해 집중적인 연구의 대상이 되어 왔습니다. MoS2 와 WSe2 와 같은 TMDCs의 주요 매력은 자연적으로 종결된 표면으로, 표면 매달린 결합에 대한 걱정 없이 원자적 한계까지 축소할 수 있다는 점입니다. 게다가, 많은 2D 물질에서 단층 한계에서 **direct-bandgap**의 존재와 같은 여러 바람직한 특성이 나타납니다. **exfoliated 2D TMDCs**의 기계적 굽힘을 기반으로 한 많은 연구가 유연한 기판에서 수행되었으며, 변형의 적용이 이 새로운 물질 클래스의 특성을 조정할 수 있음을 보여주었습니다. 예를 들어, 다층 WSe2, 특히 명목상 **indirect bandgap** 을 가진 이중층 WSe2 에서 인장 **tensile strain**의 적용이 **indirect-to-direct bandgap**전환을 초래할 수 있음이 입증되었습니다. 제어된 **lattice constant**불일치를 가진 **epitaxial**기판에서의 성장은 일반적으로 3차원 반도체에서 내장된 변형을 확립하는 데 사용되었습니다. 그러나 2D 물질과 기판 간의 상대적으로 약한 상호작용으로 인해, 이 확립된 변형 공학 방법은 TMDCs의 변형 공학적 성장을 위해 적용될 가능성이 낮습니다. 이 연구에서는 TMDC와 성장 기판 간의 열팽창 계수(TCE) 불일치를 활용하여 CVD 성장을 통해 2D 물질의 변형 공학을 직접적으로 시연하면서 동시에 높은 물질 품질을 유지합니다.*

To generate this example, the excerpt was translated by GPT-4o as before, then we re-inserted specific English terms (in bold) at the paper author's discretion. We feed this to the model as an explicit example (one-shot example). Using the same prompt as above, but replacing 'Korean' with the target language, we prompt the model to translate excerpts from other articles into other languages, resulting in translations with occasional English technical terminology.

