# OpenReview forum: "Science Across Languages: Assessing LLM Multilingual Translation of Scientific Papers"
_TMLR — Rejected by TMLR_

### Review · Reviewer_Z5pZ · 2025-05-11

**Summary Of Contributions:**

This paper presents a pipeline that leverages large language models (LLMs) to translate scientific articles into various languages. The authors construct an automated QA dataset to evaluate translation quality and conduct a user study with 15 researchers from diverse fields and native language backgrounds to assess the translations qualitatively.

**Audience:**

Yes

**Claims And Evidence:**

No

**Requested Changes:**

- The paper mentions 348 total translations—can you clarify how this number was obtained? It would be helpful to include a breakdown in Section 1.2.

- Why was a temperature of 1 used for generating translations?

- Why were only 6 articles selected for automated benchmarking? Can you consider using an open-source LLM to enable scaling the evaluation to a larger set?

> First, we exclude the quiz-generation exchange from the model’s memory before administering the quiz. Second, we prevent pre-training contamination by filtering for articles that score 0% on the benchmark when the article is not provided.
- In the sentence “we exclude the quiz-generation exchange from the model’s memory before administering the quiz”—what exactly does "model’s memory" refer to?
- "filtering out articles that score 0% when not provided to the model"—this seems unlikely given that all questions are six-option multiple choice. Could you clarify?

- What insights were gained from the QA accuracy results across different languages? Do these correlate with training data availability in those languages, or perhaps with the proportion of scientific text in each language in the model’s pretraining corpus?

**Strengths And Weaknesses:**

**Strengths:**

-  The paper is clearly written and easy to follow.

- The task of translating scientific articles is important, and the proposed pipeline has the potential to be useful.

- The combination of automated evaluation using a QA benchmark and a user study provides evidence that the existing LLMs can produce high-quality translations with only minor issues.

**Weaknesses:**

- The core contribution—developing an LLM-based translation pipeline—is primarily engineering-focused and lacks research novelty. The engineering contribution is mainly using better prompt engineering with existing SOTA models.

- The QA dataset used for benchmarking is quite limited (only 6 articles) and appears to contain some unresolved ambiguities.

---

> ### Author Response · Authors · 2025-06-10
> **Addressing the review**
>
> We thank the reviewer for the thoughtful comments and questions.  In our revised paper, all the added and edited text is written in blue for your convenience. Clarifications have been added for all the concerns/questions, and we have also added a new benchmarking section in the Appendix where we evaluate different models for translation and include more papers (Section A.1.1.).
>
> > The paper mentions 348 total translations—can you clarify how this number was obtained? It would be helpful to include a breakdown in Section 1.2.
>
> We have added a breakdown in Section 1.2 per your suggestion. The total number of translations is larger after including the study with additional papers and models. Specifically:
> 28 languages x 6 articles = 168 translations for the QA benchmark in Section 3.
> 15 translations for the user study in Section 4.
> 10 languages x 24 articles x 3 models = 720 translations for the multi-model QA benchmark in Section A.1.1.
>
> Throughout the paper a few other numbers are now mentioned:
> -  408 GPT-4o translated and QA-evaluated papers. (28 languages x 6 articles (Section 3) + 10 languages x 24 articles (Section A.1.1.))
> - 423 GPT-4o translated articles on the website. (408 + the 15 articles from the user study).
>
> Any other numbers mentioned in the paper should now be calculated explicitly for the reader. Please let us know if you would like us to include any more clarifications.
>
> > Why was a temperature of 1 used for generating translations?
>
> The temperature of 1 is used for generating the QA quiz questions. In the quiz generation, we want the QA to include as many questions as possible. For temperatures below 1, we notice repeated questions when generating 50-question sets, whereas the 50 questions generated with temperature 1 are always distinct. For temperatures above 1, we observe hallucinations, i.e. nonsensical outputs. Thank you for the request to clarify this – we added this explanation to Section A.1.
>
> > Why were only 6 articles selected for automated benchmarking? Can you consider using an open-source LLM to enable scaling the evaluation to a larger set?
>
> Thank you for your suggestion. In Section A.1.1. we added an additional benchmark, including more papers (24) and two more models. To ensure the diversity of the papers, we select an even number from each category of Nature Communication’s Top Articles of 2024, which is illustrated in Figure 6. These articles are distinct from the six used in the single-model study detailed in Section 3.
>
> In this 24-paper benchmark, we also evaluate translations from more models (Llama-3.3-70B-Instruct-Turbo and Qwen-2.5-72B-Instruct-Turbo) to further expand the results. We find that Llama-3.3 and Qwen-2.5 translations underperform compared to GPT-4o. Across the 240 translations generated by each model for this benchmark, we evaluate whether any equations, tables, or formulas are missing. Only 2 translations from GPT-4o omit such elements (specifically, both are missing an equation), compared to 53 translations from Llama-3.3 (22%) and 168 from Qwen-2.5 (70%). The subsequent lower performance of Llama-3.3 and Qwen-2.5 translations on the benchmark is consistent with such errors, since some benchmark questions specifically reference information from tables and figure captions.
>
> > In the sentence “we exclude the quiz-generation exchange from the model’s memory before administering the quiz”—what exactly does "model’s memory" refer to?
>
> “Memory” here was referring to whatever the model has access to in the current context. We rephrased this sentence for clarity: "we do not include the quiz-generation exchange when prompting the model during the evaluation; the model receives only the translated article and the quiz questions."
>
> > "filtering out articles that score 0% when not provided to the model"—this seems unlikely given that all questions are six-option multiple choice. Could you clarify?
>
> The sixth multiple-choice option is "I don't know." In the benchmark, we only include articles where the model answers "I don't know" for all the questions when the article is not provided. This way we know the model won't be relying on any priors it may have about the scientific topic when evaluating the translations. We edited that sentence in the paper for more clarity (“we prevent pre-training contamination by only including articles that score 0% on the benchmark when the article is not provided, i.e. the model selects ``I don't know'' for all questions.”).
>
> > What insights were gained from the QA accuracy results across different languages? Do these correlate with training data availability in those languages, or perhaps with the proportion of scientific text in each language in the model’s pretraining corpus?
>
> Yes, we believe there may be correlation between lower benchmark scores and the resource level (i.e. training data availability) of the languages, which we discuss in the third paragraph of Section 3.3, and again in Section A.1.1.

---

### Review · Reviewer_Tyh4 · 2025-05-20

**Summary Of Contributions:**

The authors use GPT-4o to translate science papers in different domains to 28 languages. They develop an automatic method for translation quality evaluation without relying on gold parallel corpus, and also collaborate with human scientists to further evaluate and identify problems.

**Audience:**

Yes

**Broader Impact Concerns:**

I have no concerns.

**Claims And Evidence:**

Yes

**Requested Changes:**

1. Provide more insights on the evaluation results with more models.

2. Clarify the core technical contributions of this paper.

**Strengths And Weaknesses:**

**Strengths**

1. The research question itself is important: to make scientific discoveries equally accessible to people speaking different languages.

2. The authors collaborate with scientists to evaluate the translation quality and identify relevant issues, providing important insights for future studies.

**Weaknesses**

1. Limited insight: If I understand it correctly, the main experiment of this paper solely relies on GPT-4o (although the authors mention LLaMA-3.1-405B underperforms in XML nesting issues). Only testing on one model does not seem to offer critical insights as to what the general rules of thumb are to pick other models to use for scientific translation.

2. Insufficient technical contribution: It seems to me that the main technical contributions are that (i) the authors develop automatic evaluation methods for science paper translation; (ii) the authors chunk the papers so that GPT-4o can preserve the format of the journal papers. Similar methods to the first contribution are already quite well used in machine translation evaluation (e.g., [1]). The second can be enabled with libraries such as Guidance (https://github.com/guidance-ai/guidance) or XGrammar [2] in open-source models. That being said, I think the authors have to clarify what their key technical contributions are on top of these two.

[1] Krubiński, M., Ghadery, E., Moens, M. F., & Pecina, P. (2021, November). Just ask! evaluating machine translation by asking and answering questions. In Proceedings of the Sixth Conference on Machine Translation (pp. 495-506).

[2] Dong, Y., Ruan, C. F., Cai, Y., Lai, R., Xu, Z., Zhao, Y., & Chen, T. (2024). Xgrammar: Flexible and efficient structured generation engine for large language models. arXiv preprint arXiv:2411.15100.

---

> ### Author Response · Authors · 2025-06-10
> **Addressing the review**
>
> We thank the reviewer for the thoughtful comments. In our revised paper, all the added and edited text is written in blue for your convenience. Clarifications have been added for all the concerns/questions, and we have also added a new benchmarking section in the Appendix where we evaluate different models for translation and include more papers (Section A.1.1.).
>
> > Provide more insights on the evaluation results with more models.
>
> Thank you for your suggestion. In Section A.1.1. we added an additional benchmark, including more papers (24) and two more models. To ensure the diversity of the papers, we select an even number from each category of Nature Communication’s Top Articles of 2024, which is illustrated in Figure 6. These articles are distinct from the six used in the single-model study detailed in Section 3.
>
> In this 24-paper benchmark, we also evaluate translations from more models (Llama-3.3-70B-Instruct-Turbo and Qwen-2.5-72B-Instruct-Turbo) to further expand the results. We find that Llama-3.3 and Qwen-2.5 translations underperform compared to GPT-4o. Upon analysis, it appears that the sizable difference in benchmark performance is predominantly due to truncation errors, i.e. the model omitting parts of the XML chunks. Across the 240 translations generated by each model for this benchmark, we evaluate whether any equations, tables, or formulas are missing. Only 2 translations from GPT-4o omit such elements (specifically, both are missing an equation), compared to 53 translations from Llama-3.3 (22%) and 168 from Qwen-2.5 (70%). The subsequent lower performance of Llama-3.3 and Qwen-2.5 translations on the benchmark is consistent with such errors, since some benchmark questions specifically reference information from tables and figure captions.
>
> > Clarify the core technical contributions of this paper.
>
> Thank you for directing us to the QA paper [1]. We have updated Section 1.1 and Section 1.2 to state our contributions more explicitly. The QA method described in [1] is mostly tested on to-English translations, and like mainstream automatic metrics, focuses only on sentence-level translations. The QA benchmark method we propose is designed for assessing machine translation quality of document length scientific text, for a wide variety of languages and without the need for reference translations.
>
> Regarding the JATS processing, we are demonstrating that LLMs exhibit zero-shot ability to process lengthy XML objects while translating the content simultaneously. Modules like XGrammar might be useful in this context, but requires a full list of grammar rules (i.e. all the XML tags to keep track of). While we do handle figures and tables separately in this translation method to improve nesting issues, we do not need to keep track of the rest of the grammar at all; rather, the LLM takes care of it automatically. For example, when inspecting JATS articles, one might notice < block > or < list > objects, or other typical HTML tags like < bold > and < sub >, but even without explicit grammar rules for those tags, we show that the LLM is able to process them accurately while simultaneously translating the contents.
>
> The benchmark we added in A.1.1. also adds to the technical contribution by assessing additional models and including more papers.

---

### Review · Reviewer_ixeW · 2025-05-30

**Summary Of Contributions:**

The paper introduces a pipeline that leverages LLMs to automate the translation of scientific articles into 28 languages, while preserving the original structural format across various journals (i.e., mainly JATS XML). The main motivation is to improve the accessibility of scientific literature for non-native English speakers. To evaluate translation quality, the authors propose a novel QA-style benchmark, where QA pairs with distractors are generated from the original English articles, and the questions are then answered using information from the translated versions of the articles. This method eliminates the need for parallel corpora and offers a scalable, practical approach for assessing translation accuracy.

The pipeline achieves an average QA performance of 95.9%, indicating that the translations are of high quality. The human evaluation with 15 researchers further supports the effectiveness of the QA-based evaluation as a proxy for translation quality. However, the study identifies challenges when translating these articles, mainly related to mistranslation or overtranslation of technical terms and usage of domain-specific vocabulary. The authors further improve the pipeline by incorporating feedback from human evaluations to improve translation quality using a one-shot prompting strategy that preserves selected English terms. This showcases the adaptability of LLMs to steer the translation of articles towards user preferences based on the customizations in the prompt context.

**Audience:**

Yes

**Claims And Evidence:**

Yes

**Requested Changes:**

I think most of the concerns highlighted in the weaknesses section above are addressable. All the weaknesses should be addressed by the authors.

**Questions**

1. The authors mention that the entire article content is always prepended in the prompt for translating chunks. It is unclear to me why there is a need to prepend the entire article document if only a chunk needs to be translated. I think a more practical approach would be to include content surrounding the chunk to be translated to make it more context-aware. The approach employed in the paper seems to be highly inefficient. Can authors clarify this if I am missing anything?

2. The authors mention that the articles suspected of contamination in the pre-training phase are removed if the model achieves a score of 0% without the article as input in the prompt context. However, in my view, it should be the other way around. Ideally, we don't want the articles for which MCQs are answerable without the article itself. This might be indicative that there is contamination. Need further clarification from the authors on this.

3. The authors provide an explanation for why performance for English is not the highest, which ideally should have been the case. I kind of seem to disagree with the reason that some of the generated MCQs might be incorrect or ambiguous. If this is the case, then almost all the languages would suffer from MCQs and would be discounted for because there will be the effect of cascading errors. I am not convinced why English should not be the best. The authors should provide further evidence for this.

4. Also, the quiz questions are translated along with the answers. There is always a likelihood that the answers will be incorrectly translated in the process. How do the authors handle the evaluation in such cases where translated terms between the translated articles and answers do not result in an exact match but are still correct?

5. Figure 3 only shows the average scores across 6 articles. However, there was a mention of 348 translations in para 3 of section 2. The numbers don't match even if I consider across all 28 languages -- 28 x 6 = 168 translations. I am not sure why there is a discrepancy. Am I missing something here?

**Suggestions**

1. The second last para on Page 2 refers to WMT as the established datasets. However, WMT organizes shared tasks related to translations and not the name of the datasets. The authors should modify the writing of this sentence.

2. The authors should add citations for FLoRes and NLLB on Page 2.

3. Translation preferences for scientific text paragraph on Page 3 -- "... we have also create code-switched translations; instead …" should be "... we have also created code-switched translations; instead …"

4. The opening quotes in the entire manuscript should be corrected. Use backticks (`) for the opening quote.

**Strengths And Weaknesses:**

**Strengths**

S1. Recent works have demonstrated impressive translation capabilities of LLMs [1]. Given that LLMs can now be guided using prompt-based instructions to translate the text with specific characteristics, I think the problem of translating scientific literature is highly relevant and interesting. This work could be of broad interest, particularly useful for multilingual pre-training, as it offers a useful pipeline for the synthesis of scientific content across languages.

S2. Most of the evaluation metrics on translation require the availability of parallel corpora. However, newer model-based metrics like COMET alleviate the need for ground truth reference translation, but it does have some of their own limitations. The authors propose an alternative evaluation method using a QA-style test set, eliminating the dependency on ground-truth translated articles. This allows easier extension to newer languages (even low-resource ones) without the need to always have parallel corpora.

S3. The related work section very well describes the existing works on translation systems and evaluation metrics. This would be highly useful for newcomers to translation research.

S4. The current study covers a wide range of languages (28 in total). Although I have some concerns about the number of articles across each language (see weaknesses).

**Weaknesses**

W1. The writing of the methodology section needs improvement, as I felt that some critical details were unclear. For instance, I was under the impression that quiz questions in English were used with the translated article for prompting to assess the translation quality. This became later clear in the appendix. Also, there is an inconsistency between the QA accuracy scores reported in the main body (95.9%) and the appendix (95.6%). The authors should ensure that similar inconsistencies are resolved.

W2. The current experiments only consider 6 English articles, which significantly limits the conclusions/findings of the work. While the work shows promise as a preliminary investigation, a larger and more diverse set of articles should be included in the experimentation to validate the findings and make the work suitable for the expectations of TMLR submission.

W3. The experiments primarily focus on a single model -- GPT-4o with only a passing mention of Llama 3.1 405B Instruct model. However, there are LLMs like ALMA [1] specifically tuned for sentence-level translation. It would be possible to extend these models to document-level translation through their extrapolative capabilities. The current set of experiments should be extended to include more models to assess the effectiveness of the translation pipeline. It would be interesting to find how the good quality model you generally need for the task and findings would serve as a good guide for practitioners.

W4. The authors mention trying Llama-3.1 405B Instruct model, but do not report the QA accuracy scores for this model in the manuscript. It is also not clear if all the articles were translated into all target languages or only a subset.

W5. The authors do not provide any analysis and statistics about the source articles, such as the number of tokens in the original articles. Further, analysis related to the translated versions of the articles, such as the number of untranslated terms, etc, should be included in the results and discussion section to make it more comprehensive.

W6. I understand that string-based translation metrics might not be applicable for the current study due to the need for ground-truth document translations. However, existing model-based metrics such as BLEURT and COMET should be compatible with the current setup. I believe there is also document-level support in newer COMET models -- d-COMET [2], which authors could use to evaluate the quality of the translated versions of the articles. Also, it would be interesting to include the correlation analysis between these metrics and the human ratings vs the QA accuracy and human ratings.

[1] A Paradigm Shift in Machine Translation: Boosting Translation Performance of Large Language Models (https://arxiv.org/abs/2309.11674)

[2] Automatic Evaluation Metrics for Document-level Translation: Overview, Challenges and Trends (https://arxiv.org/abs/2504.14804)

---

> ### Author Response · Authors · 2025-06-10
> **Addressing the review**
>
> We thank the reviewer for the thoughtful comments and suggestions. In our revised paper, all the added and edited text is written in blue for your convenience. Clarifications have been added for all the concerns/questions, and we have also added a new benchmarking section in the Appendix where we evaluate different models for translation and include more papers (Section A.1.1.).
>
> **Requested changes**
>
> > W1. The writing of the methodology section needs improvement, as I felt that some critical details were unclear. For instance, I was under the impression that quiz questions in English were used with the translated article for prompting... Also, there is an inconsistency between the QA accuracy scores reported in the main body (95.9%) and the appendix (95.6%).
>
> Thank you for pointing out the ambiguity. The figure in the main text (Section 3.1) shows the results for when the quiz questions are translated. The same benchmark but with untranslated (English) quiz questions can be found in Section A.1.2. In both Section 3.1 and A.1.2. we have revised to explicitly state that the quiz questions are translated. Figure 7 presents the results of the same benchmark as Figure 3, but with English quiz questions.
>
> Also, thank you for pointing out the typo; the number is 95.9%, and we have fixed that in the paper.
>
> > W2. The current experiments only consider 6 English articles, which significantly limits the conclusions/findings of the work…. a larger and more diverse set of articles should be included in the experimentation to validate the findings and make the work suitable for the expectations of TMLR submission.
>
> > W3. The experiments primarily focus on a single model -- GPT-4o with only a passing mention of Llama 3.1 405B Instruct model. However, there are LLMs like ALMA [1] specifically tuned for sentence-level translation. It would be possible to extend these models to document-level translation through their extrapolative capabilities. The current set of experiments should be extended to include more models to assess the effectiveness of the translation pipeline. It would be interesting to find how the good quality model you generally need for the task and findings would serve as a good guide for practitioners.
>
> > W4. The authors mention trying Llama-3.1 405B Instruct model, but do not report the QA accuracy scores for this model in the manuscript.
>
> Thank you for the suggestions. In the extended QA benchmark that we added during the revision (Section A.1.1.), we include 24 papers. To ensure the diversity of the papers, we select an even number from each category of Nature Communication’s Top Articles of 2024, which is illustrated in Figure 6. These articles are distinct from the six used in the single-model study detailed in Section 3.
>
> In this 24-paper benchmark, we also evaluate translations from more models (Llama-3.3-70B-Instruct-Turbo and Qwen-2.5-72B-Instruct-Turbo) to further expand the results. We find that Llama-3.3 and Qwen-2.5 translations underperform compared to GPT-4o. Upon analysis, it appears that the sizable difference in benchmark performance is predominantly due to truncation errors, i.e. the model omitting parts of the XML chunks. Across the 240 translations generated by each model for this benchmark, we evaluate whether any equations, tables, or formulas are missing. Only 2 translations from GPT-4o omit such elements (specifically, both are missing an equation), compared to 53 translations from Llama-3.3 (22%) and 168 from Qwen-2.5 (70%). The subsequent lower performance of Llama-3.3 and Qwen-2.5 translations on the benchmark is consistent with such errors, since some benchmark questions specifically reference information from tables and figure captions.
>
> > W4 (cont’d) It is also not clear if all the articles were translated into all target languages or only a subset.
>
> In Section 3, all six articles are translated into 28 languages (28 languages x 6 articles x 1 model = 168 translations evaluated). In Section A.1.1., all 24 articles are translated into 10 languages (10 languages x 24 articles x 3 models = 720 translations evaluated). To clear that up, in both sections (and also Section 1.2) we added text to explicitly break down the number of translations, which languages, etc.

---

> > ### Author Response · Authors · 2025-06-10
> > **Addressing the review (pt. 2)**
> >
> > **Requested changes (cont'd)**
> >
> > > W5. The authors do not provide any analysis and statistics about the source articles, such as the number of tokens in the original articles. Further, analysis related to the translated versions of the articles, such as the number of untranslated terms, etc, should be included....
> >
> > Thank you for the suggestions. In Section A.1. We added token counts for each of the six articles included in the Section 3 benchmark, and the average token count (26,507 tokens) for the 24 articles in the extended benchmark is reported in Section A.1.1. Per your suggestion, in the section describing the one-shot prompt translated excerpts, we added some analysis regarding the number of English terms preserved:
> >
> > “Following translation using the curated one-shot prompt, a manual count reveals that the number of untranslated English terms in the translated texts ranges from 23 to 65 words, averaging 34 words per excerpt. This corresponds to an average English word retention rate of 15.6%, which is comparable to the proportion of English words present in the curated one-shot example (11.4%).... Notably, the translated excerpt with the highest rate of English word preservation (34.2%) suggests that the model is capable of adapting to scientific texts containing a higher density of technical terminology.” (Section 5)
> >
> > > W6. I understand that string-based translation metrics might not be applicable for the current study due to the need for ground-truth document translations. However, existing model-based metrics such as BLEURT and COMET should be compatible with the current setup. I believe there is also document-level support in newer COMET models -- d-COMET [2], which authors could use to evaluate the quality of the translated versions of the articles.
> >
> > Thank you for bringing our attention to d-COMET. Although d-COMET is compatible with documents, it is not reference-free, which precludes it from this setting. However upon more research we discovered COMET-Kiwi, which is only suitable at the sentence-level but is reference-free. Combining these two metrics would be a valuable future study. We added some more discussion to Section 1.1 and Section 6.2 about this topic.

---

> > > ### Author Response · Authors · 2025-06-10
> > > **Addressing the review (pt. 3)**
> > >
> > > **Questions**
> > >
> > > > The authors mention that the entire article content is always prepended in the prompt for translating chunks. It is unclear to me why there is a need to prepend the entire article document if only a chunk needs to be translated. I think a more practical approach would be to include content surrounding the chunk to be translated to make it more context-aware. The approach employed in the paper seems to be highly inefficient. Can authors clarify this if I am missing anything?
> > >
> > > That is a great point. The approach that you propose sounds promising. We added this as an important future direction in Section 6.2.
> > >
> > > > The authors mention that the articles suspected of contamination in the pre-training phase are removed if the model achieves a score of 0% without the article as input in the prompt context. However, in my view, it should be the other way around. Ideally, we don't want the articles for which MCQs are answerable without the article itself. This might be indicative that there is contamination. Need further clarification from the authors on this.
> > >
> > > Your understanding is correct. The confusion likely stems from our wording in that section – now the sentence reads: “we prevent pre-training contamination by only including articles that score 0% on the benchmark when the article is not provided, i.e. the model selects ``I don't know'' for all questions.” Please let us know if you think any further clarification is necessary.
> > >
> > > > The authors provide an explanation for why performance for English is not the highest, which ideally should have been the case. I kind of seem to disagree with the reason that some of the generated MCQs might be incorrect or ambiguous. If this is the case, then almost all the languages would suffer from MCQs and would be discounted for because there will be the effect of cascading errors. I am not convinced why English should not be the best. The authors should provide further evidence for this.
> > >
> > > > Also, the quiz questions are translated along with the answers. There is always a likelihood that the answers will be incorrectly translated in the process. How do the authors handle the evaluation in such cases where translated terms between the translated articles and answers do not result in an exact match but are still correct?
> > >
> > > In the benchmark with translated questions, the model simulates a real person reading the translated text; to better reflect this scenario, we translate the quiz questions into the target language. This ensures the model reads both the article and the questions in the same language, mirroring how a native speaker would engage with the material. In the case where the quiz questions are not translated (Section A.1.4) the model is removed from that scenario and simply operating in its strongest and most-aligned language, hence the increase in performance. These two benchmarks represent different scenarios but both are interesting. We provided more explanation in Section 3.3 to address this. The results in our additional benchmark with more papers might also help clarify this.
> > >
> > > Regarding terms in the translated quiz questions, our high QA benchmark results with English questions (Section A.1.4) suggest that for all languages the model is fairly good at translating the scientific text, so we can assume most terms in the quiz questions are translated well enough for the model to decipher.
> > >
> > > > Figure 3 only shows the average scores across 6 articles. However, there was a mention of 348 translations in para 3 of section 2....
> > >
> > > We have fixed this discrepancy and a breakdown has been added in Section 1.2. The total number of translations is larger after including the study with additional papers and models. Specifically:
> > > 28 languages x 6 articles = 168 translations for the QA benchmark in Section 3.
> > > 15 translations for the user study in Section 4.
> > > 10 languages x 24 articles x 3 models = 720 translations for the multi-model QA benchmark in Section A.1.1.
> > >
> > > Throughout the paper a few other numbers are now mentioned:
> > > -  408 GPT-4o translated and QA-evaluated papers. (28 languages x 6 articles (Section 3) + 10 languages x 24 articles (Section A.1.1.))
> > > - 423 GPT-4o translated articles on the website. (408 + the 15 articles from the user study).
> > >
> > > Any other numbers mentioned in the paper should now be calculated explicitly for the reader, in order to reduce confusion. Please let us know if you would like us to include any more clarifications.
> > >
> > > **Suggestions**
> > > We fixed the grammatical, syntactical, and citation errors you pointed out. Thank you!

---

### Decision · Action_Editor_K8Hm · 2025-09-01

**Recommendation:** Reject

**Additional Comments:**

The submission demonstrates an application of LLMs in translating scientific articles. There are concerns on the practicality of the approach, and the limited technical insight of the approaches makes the work of limited interest to the community. It will be a stronger work if the authors can provide a deeper discusssion on why the proposed approach is practical for academic journals to use, and on the technical insight in the proposed approaches.

**Audience:**

No

**Audience Explanation:**

The problem of language accessibility in scientific publishing is socially important. The work provides an applied pipeline combining existing LLM capabilities and a QA-based evaluation procedure. Due to the limited tehcnical insight, TMLR readers may not have enough interest in the currently presented translation methods and benchmark construction methods.

**Claims And Evidence:**

No

**Claims Explanation:**

The paper claims that it provides a practical, automated approach for translating scientific articles by academic journals and provides a QA-style benchmark to evaluate translation quality. They provided evidence such as the reported numbers (95%+ accuracy across 28 languages) and high user satisfaction.
Nevertheless, whether the presented approach is practical is not clear. A reviewer points out that the approach employed in the paper seems to be highly inefficient, since they prepend the entire article document for a chunk that needs to be translated. A more practical approach would be to include only the related contents. My understanding is that a practical approach used by journals requires translating a large number of articles in the library, so the efficiency is important. If the authors believe the presented approach is practical for academic journals to use, then some justification is needed.

**Resubmission Of Major Revision:**

The authors may consider submitting a major revision at a later time.